

# How do you feel, developer? An explanatory theory of the impact of affects on programming performance

Daniel Graziotin[1], Xiaofeng Wang[1] and Pekka Abrahamsson[2]

[1] Faculty of Computer Science, Free University of Bozen-Bolzano, Bolzano/Bozen, Italy
[2] Department of Computer and Information Science, Norwegian University of Science and Technology, Trondheim, Norway

## ABSTRACT

Affects—emotions and moods—have an impact on cognitive activities and the working performance of individuals. Development tasks are undertaken through cognitive processes, yet software engineering research lacks theory on affects and their impact on software development activities. In this paper, we report on an interpretive study aimed at broadening our understanding of the psychology of programming in terms of the experience of affects while programming, and the impact of affects on programming performance. We conducted a qualitative interpretive study based on: face-to-face open-ended interviews, in-field observations, and e-mail exchanges. This enabled us to construct a novel explanatory theory of the impact of affects on development performance. The theory is explicated using an established taxonomy framework. The proposed theory builds upon the concepts of events, affects, attractors, focus, goals, and performance. Theoretical and practical implications are given.

## INTRODUCTION

It has been established that software development is intellectual, and it is carried out through cognitive processes (*Feldt et al., 2010*; *Feldt et al., 2008*; *Khan, Brinkman & Hierons, 2010*; *Lenberg, Feldt & Wallgren, 2014*; *Lenberg, Feldt & Wallgren, 2015*). Software development happens in our minds first, then on artifacts (*Fischer, 1987*). We are human beings, and, as such, we behave based on affects as we encounter the world through them (*Ciborra, 2002*). Affects—which for us are emotions and moods[1] —are the medium within which acting towards the world takes place (*Ciborra, 2002*).

The affects pervade organizations because they influence worker's thoughts and actions (*Brief & Weiss, 2002*). Affects have a role in the relationships between workers, deadlines, work motivation, sense-making, and human-resource processes

Corresponding author
Daniel Graziotin,
daniel.graziotin@unibz.it

---

[1] For the purposes of this study, we consider affect as an underlying term for emotions and moods, in line with several other authors, e.g., *Weiss & Cropanzano (1996)*; *Fisher (2000)*. See "Affect, emotion, and mood" for more information.

(*Barsade & Gibson, 2007*). Although affects have been historically neglected in studies of industrial and organizational psychology (*Muchinsky, 2000*), an interest in the role of affects on job outcomes has accelerated over the past fifteen years in psychology research (*Fisher & Ashkanasy, 2000*). In particular, the link between affects and work-related achievements, including performance (*Barsade & Gibson, 2007*; *Miner & Glomb, 2010*; *Shockley et al., 2012*) and problem-solving processes, such as creativity (*Amabile et al., 2005*; *Amabile, 1996*), has been of interest for recent research. While research is still needed on the impact of affects to cognitive activities and work-related achievements in general, this link undeniably exists according to psychology research. We believe that it is important to understand the role of affects in software development processes and their impact on the performance[2] of developers.

It has been argued that software engineering has to produce knowledge that matters to practitioners (*Osterweil et al., 2008*). Indeed, we have shown elsewhere (*Graziotin, Wang & Abrahamsson, 2014b*) that practitioners are deeply interested in their affects while developing software, which causes them to engage in long and interesting discussions when reading related articles.

We share *Lenberg, Feldt & Wallgren (2015)* view that software engineering should also be studied from a behavioral perspective. We have embraced this view in previous studies—e.g., *Graziotin, Wang & Abrahamsson (2014a)*, *Graziotin, Wang & Abrahamsson (2015a)* and have employed theories and measurement instruments from psychology to understand how affect impact o software developers' performance under a quantitative strategy using experiments. However, in order to understand the human behavior behind affects and software development, there is a need to observe software developers in-action and perform interviews. So far, research has not produced qualitative insights on the mechanism behind the impact of affects on the performance of developers. We have called for such studies in the past (*Graziotin, Wang & Abrahamsson, 2015a*). Moreover, a lack of theory in software engineering has been recently highlighted (*Johnson, Ekstedt & Jacobson, 2012*).

Thus, we conducted a study laying down the theoretical answers to the research question *how are developers' experienced affects related to performance while programming?* In this paper, we report an interpretive study of the impact of affects of developers on the software development performance. By deeply observing and open interviewing two developers during a development cycle, we constructed an explanatory theory, called *Type II* theory by *Gregor (2006)*, for explaining the impact of affects on development performance.

The remainder of this paper is structured as follows. In the *Background* section, we first briefly introduce what we mean with *affects*. We then review the related studies of affects and the performance of developers. Then, we provide the theoretical framing of this study and the theory representation. The following section summarizes the methodology of this study by explicating our worldview and how we chose among the various options, the research design, the data analysis method, and the reliability procedures. We then report the results of our work, i.e., an explanatory theory of the impact of affects on programming performance, as well as a discussion and comparison with related work. The last section

[2] The stance that performance and productivity are two interchangeable terms is assumed in this study, in line with *Fagerholm et al. (2015)*, *Petersen (2011)* and *Meyer et al. (2014)*.

concludes the paper by providing the contribution and implications of our study, the limitations, and the suggested future work.

## BACKGROUND

In this section, we first briefly introduce what we mean with *affects*, and we review the papers in the software engineering field, where the affects of software developers have been taken into consideration with respect to performance.

### Affect, emotion, and mood

The fields of psychology have yet to agree on the definitions of affects and the related terms such as emotions, moods, and feelings (*Ortony, Clore & Collins, 1990*; *Russell, 2003*). Several definitions for affects, emotions, and moods exist—to the point that *Ortony, Clore & Collins (1990)* defined the study of affects as a "very confused and confusing field of study" (p. 2). We are aware that some proposals have been established more than others. For example, *Plutchik & Kellerman (1980)* have defined *emotions* as the states of mind that are raised by external stimuli and are directed toward the stimulus in the environment by which they are raised. *Parkinson et al. (1996)* have defined *moods* as emotional states in which the individual feels good or bad, and either likes or dislikes what is happening around him/her. In other words, mood has been defined as a suffused emotion, where no originating stimulus or a target object can be distinguished (*Russell, 2003*).

The issue with the proposed definitions, including those reported above, is that hundreds of competing definitions have been produced in just a few years (*Kleinginna & Kleinginna, 1981*) and a consensus has yet to be reached. There are also cultural issues to be considered. For example, emotion as a term is not universally employed, as it does not exist in all languages and cultures (*Russell, 1991*). Distinctions between emotions and moods are clouded, because both may feel very much the same from the perspective of an individual experiencing either (*Beedie, Terry & Lane, 2005*).

As emotions and moods may feel the same from the perspective of an individual, we have adopted the stance of several researchers in the various fields (*Schwarz & Clore, 1983*; *Schwarz, 1990*; *Wegge et al., 2006*; *De Dreu et al., 2011*) and employed the noun *affects* (and affective states) as an underlying term for emotions and moods. We do not neglect moods and emotions *per se*.[3] We opted to understand the states of minds of software developers at the *affective* level only, that is "one level below" moods and emotions. Our choice was not unthoughtful. We have adhered to the *core affect* theory (*Russell, 2003*; *Russell & Barrett, 1999*; *Russell, 2009*), which employs affect as the atomic unit upon which moods and emotional experiences can be constructed. That is, in this article we do not distinguish between emotions and moods. We are interested in understanding how developers feel.

### Related work

*Lesiuk (2005)* studied 56 software engineers in a field study with removed treatment design.[4] The aim of the study was to understand the impact of music listening on software design performance. The study was conducted over a five-week period. The design performance and the affects of the developers were self-assessed twice per day. For the first

---

[3] The issues of defining the concepts under study is not trivial and it deserves separate discussions. We point the reader to two of our recent articles (*Graziotin, Wang & Abrahamsson, 2015c*; *Graziotin, Wang & Abrahamsson, 2015b*), in which we have discussed the theoretical foundations, the various theories, and the classification frameworks for affects, emotions, and moods, and the common misconceptions that occur when studying these constructs.

[4] Removed treatment designs are part of single-group quasi-experiment designs. A removed treatment design allows one to test hypotheses about an outcome in the presence of the intervention and in the absence of the intervention (*Harris et al., 2006*). A pre-treatment measurement is taken on a desired outcome; a treatment is provided; a post-treatment measurement is conducted; a second post-treatment measurement is conducted; the treatment is removed; a final measurement is performed (*Harris et al., 2006*).

week of the study (the baseline), the participants were observed in natural settings—that is, they worked as usual, doing what they do usually. During the second and third week, the participants were allowed to listen to their favorite music while working. However, during the fourth week, listening to music was not allowed. During the fifth week, the participants were allowed again to listen to the music. The results indicated a positive correlation of positive affects and listening to favorite music. Positive affects of the participants and self-assessed performance were lowest with no music, but not statistically significant. On the other hand, narrative responses revealed the value of music listening for positive mood change and enhanced perception on software design performance.

Along a similar line, *Khan, Brinkman & Hierons (2010)* theoretically constructed links from psychology and cognitive science studies to software development studies. In this construction, programming tasks were linked to cognitive tasks, and cognitive tasks were linked to affects. For example, the process of constructing a program—e.g., modeling and implementation—was mapped to the cognitive tasks of memory, reasoning, and induction. *Khan, Brinkman & Hierons (2010)* conducted two studies to understand the impact of affects on the debugging performance of developers. In the first study, positive affects were induced to the software developers. Subsequently, the developers completed a quiz about software debugging. In the second study, the participants wrote traces of the execution of algorithms on paper. During the task, the affect arousal was induced to the participants. Overall, the results of the two studies provided empirical evidence for a positive correlation between the affects of software developers and their debugging performance.

We also conducted two studies to understand the connection between affects and the performance of software developers. In the first study (*Graziotin, Wang & Abrahamsson, 2014a*), we recruited 42 computer science students to investigate the relationship between the affects of software developers and their performance in terms of creativity and analytic problem-solving. In a natural experiment, the participants performed two tasks chosen from psychology research that could be transposed to development activities. The participants' pre-existing affects were measured before each task. Overall, the results showed that the happiest developers are better problem solvers in terms of their analytic abilities.

The second study (*Graziotin, Wang & Abrahamsson, 2015a*) was a correlation study of real-time affects and the self-assessed productivity of eight software developers while they were performing a 90 min programming task on a real-world project. The developers' affects and their productivity were measured in intervals of 10 min. Through the fit of a linear mixed effects model, we found evidence for a positive correlation between the affects of developers associated to a programming task and their self-assessed productivity. In this study, we called for process-based studies on software teams which "are required in order to understand the dynamics of affects and the creative performance of software teams and organizations" (p. 17).

*Müller & Fritz (2015)* performed a study with 17 participants, 6 of which were professional software developers and 11 were PhD students in computer science. The participants were asked to perform two change tasks, one for retrieving StackOverflow scores and the other to let users undo more than one command in the JHotDraw program. During the

development, the participants were observed using three biometric sensors, namely an eye tracker, an electroencephalogram, and a wearable wireless multi-sensor for physiological signals (e.g., heart rate, temperature, skin conductance). After watching a relaxing video, the participants worked on both tasks in a randomly assigned order. They were then interrupted after 5 min of working or when they showed strong signs of emotions. During each interruption, the participants rated their affects using a psychology measurement instrument. After other 30 min of work, the participants repeated the experiment design using the second task. Finally, the participants were interviewed. Overall, the study found that (1) developers feel a broad range of affects, expressed using the two dimensional measures of valence and arousal instead of labeling the affects, (2) the affects expressed as valence and arousal dimensions are correlated with the perceived progress in the task (evaluated using a 1–5 Likert scale), (3) the most important aspects that affect positive emotions and progress are the ability to locate and understand relevant code parts, and the mere act of writing code instead of doing nothing. On the other hand, most negative affects and stuck situations were raised by not having clear goals and by being distracted.

So far, the literature review has shown that the number of studies regarding the affects and the performance of developers is limited. Furthermore, the studies are all quantitative and toward variance theory.

Variance theories, as opposed to process theories, provide explanations for phenomena in terms of relationships among dependent and independent variables (*Langley, 1999*; *Mohr, 1982*). In variance theory, the precursor is both a necessary and sufficient condition to explain an outcome, and the time ordering among the independent variables is immaterial (*Pfeffer, 1983*; *Mohr, 1982*). Strictly speaking, variance theory studies are hypothesis-driven studies, which aim to quantify the relationship between two variables in their base case.

Process research is concerned with understanding *how* things evolve over time and *why* they evolve in they way we observe (*Langley, 1999*). According to *Langley (1999)*, process data consist mainly of "stories"—which are implemented using several different strategies—about what happened during observation of events, activities, choice, and people performing them, over time. *Mohr (1982)* has contrasted process theory from variance theory by stating that the basis of explanation of things is a probabilistic rearrangement instead of clear causality, and the precursor in process theory is only a necessary condition for the outcome.

In the literature review, a lack of theoretical and process-based studies was identified. For this reason, we aimed at developing a process-based theory.

## Theoretical framework

Our theoretical framework was primarily based upon the Affective Events Theory (AET) by *Weiss & Cropanzano (1996)* and the episodic process model of performance episodes by *Beal et al. (2005)*. AET has been developed as a high-level structure to guide research on how affects influence job satisfaction and job-related performance.

In AET, the work environment settings (e.g., the workplace, the salary, promotion opportunities, etc.) mediate work events that cause affective reactions, which are interpreted according to the individuals' disposition. Affective reactions then influence work-related behaviors. Work-related behaviors are divided into affect-driven behaviors and judgment-driven behaviors. Affect-driven behaviors are behaviors, decisions, and judgments that have immediate consequences of being in particular emotions and moods. One example could be overreacting to a criticism. Judgment-driven behaviors are driven by the more enduring work attitudes about the job and the organization (*Weiss & Beal, 2005*). Examples are absenteeism and leaving.

As *Weiss & Beal (2005)* noted ten years after publishing AET, AET has often been erroneously employed as a theoretical model to explain affective experiences at work. However, AET is a *macrostructure* for understanding affects, job satisfaction in the workplace, and to guide future research on what are their causes, consequences, and explanations. More specifically, AET is not a framework to explain the performance on the job, neither is it a model to explain the impact of all affects on job-related behaviors.

In their conceptual paper, *Beal et al. (2005)* provided a model that links the experiencing of affects to individual performance. *Beal et al. (2005)* model is centered around the conceptualization of performance episodes, which relies on self-regulation of attention regarding the on-task focus and the off-task focus. The cognitive resources towards the focus switch is limited. Affects, according to *Beal et al. (2005)*, hinder the on-task performance regardless of them being positive or negative. The reason is that affective experiences create cognitive demand. Therefore, affective experiences, according to this model, influence the resource allocation towards off-task demand.

## Theory construction and representation

Interpretive research is often conducted when producing theories for explaining phenomena (*Klein & Myers, 1999*). *Gregor (2006)* examined the structural nature of theories in information systems research. Gregor proposed a taxonomy to classify theories with respect to how they address the four central goals of analysis and description, explanation, prediction, and prescription. We employed the widely established *Gregor (2006)* work as a framework for classifying and expressing our proposed theory.

A *Type II*—or explanation—theory provides explanations but does not aim to predict with any precision. The structural components of a Type II theory are (1) the means of representation—e.g., words, diagrams, graphics, (2) the constructs—i.e., the phenomena of interests, (3) the statements of relationships—i.e., showing the relationships between the constructs, (4) the scope—the degree of generality of the statements of relationships (e.g., some, many, all, never) and statements of boundaries, and (5) the causal explanations which are usually included in the statements of relationship. While conducting this study, we ensured the constructed theory was composed of these elements.

Our study attempts to broaden our understanding of topics that are novel and unexplored in our field. *Rindova (2008)* warned us that "novelty, however, comes at a cost: novel things are harder to understand and, especially, to appreciate" (p. 300).

Therefore, we have to proceed carefully in the theory building process. The risk is to get lost in complex interrelated constructs in a confused and confusing field of study (*Ortony, Clore & Collins, 1990*) brought in the complicated, creative domain that is software engineering. Furthermore, *Barsade & Gibson (1998)* advised researchers that, when understanding emotion dynamics, the bigger is the team under observation, the more complex and complicated are the team dynamics. Bigger teams have complicated, and even historical, reasons that are harder to grasp—triggering a complex, powerful network of affects (*Barsade & Gibson, 1998*). Therefore, there is the need to keep the phenomenon under study as simple as possible. For novel theory development, philosophers and economists often—but not always—draw from their own personal observation and reasoning, while still being able to offer a sound empirical basis (*Yeager, 2011*). Theorizing from the ivory tower can complement the scientific method by offering insights and discovering necessary truths (*Yeager, 2011*), to be further expanded by empirical research. Our empirical stance makes us eager to jump to data and start theorizing; yet, we need to take some precautionary measures before doing this.

When novel theories are to be developed in new domains, such as software engineering, a small sample should be considered (*Järvinen, 2012*). A small sample enables the development of an in-depth understanding of the new phenomena under study (*Järvinen, 2012*) and to avoid isolation in the ivory tower. Our research follows carefully *Järvinen (2012)* recommendations, which is reflected in our study design. *Weick (1995)* classic article is of the same stance by reporting that organizational study theories are approximations of complex interrelated constructs of human nature that often have small samples. Those works are often seen as substitutes of theory studies, but they often represent "struggles in which people intentionally inch toward stronger theories" (ibid, p. 1). Such struggles are needed when a phenomenon is too complex to be captured in detail (*Weick, 1995*). These issues were taken into account when we designed our study, which is demonstrated in the following section.

## METHODOLOGY

We describe our research as a qualitative interpretive study, which was based on face-to-face open-ended interviews, in-field observations, and e-mail exchanges. Given the aim of the study, there was the need to make sense of the developers' perceptions, experiences, interpretations, and feelings. We wanted to conduct open-ended interviews where the realities constructed by the participants are analyzed and reconstructed by the researcher.

Our epistemological stance for understanding these social constructs and interactions has been interpretivism, which we make coincide with social constructivism in line with other authors (*Easterbrook et al., 2008*). Interpretive data analysis has been defined succinctly by *Geertz (1973)* as "really our own constructions of other people's constructions of what they and their compatriots are up to" (p. 9). Interpretivism is now established in information systems research (*Walsham, 2006*), but we see it still emerging in software engineering research.

## Design

As per our chosen design, the participants could be free to undergo the development of the system in any way, method, practice, and process they wished to employ. Our study comprised of regular scheduled face-to-face meetings with recorded interviews, impromptu meetings which could be called for by the participants themselves, e-mail exchanges, in-field observations, and a very short questionnaire right after each commit in the git system (explained in section *Reliability*). Therefore, the participants had to be aware of the design itself, although they were not informed about the aims of the study.

The participants' native language is Italian, but they have been certified as proficient English speakers. The first author of the present article employs Italian as first language as well, and he was the reference person for the participants for the duration of the entire study. The other two authors of the present article have been certified as proficient and upper intermediate in Italian. The choice for the design of the study was therefore to conduct the interviews in Italian, as the native language let the participants express their opinion and feelings in the richest, unfiltered way (*Van Nes et al., 2010*). The interviews were subsequently transcribed in English as suggested by the common research practices (*Van Nes et al., 2010*; *Squires, 2009*), but the present case had the added value that the authors could validate the transcripts with the participants over the course of the study, given their advanced proficiency in English.

The in-field observations were performed by two of the present authors, and the personal communications such as e-mails or some impromptu meetings were exchanged between the first author of the study and the participants. The coding activities have been a collaborative effort among all the authors of this study.

In order to keep the study design and results as simple as possible and to provide precise answers to the research question, in line with what we stated in the section *Theory Construction and Representation*, we observed activities that produced code. Other artifacts such as requirements and design were not taken into consideration. Furthermore, our strategy to limit the complex network of triggered affects was to group and study them into the two well-known dimensions of positive and negative affects (*Watson, Clark & Tellegen, 1988*), which assign the affects—including those perceived as neutral—in a continuum within the two dimensions.

Our design took into account ethical issues, starting with a written consent to be obtained before starting any research activity. The consent form informed the participants of our study in terms of our presence, activities, data recordings, anonymity and data protection, and that their voluntary participation could be interrupted at any time without consequences. They were also informed that any report of the study had to be approved by them in terms of their privacy, dignity protection, and data reliability before it was disclosed to any third party. Furthermore, as an extra measure, any additional, personal data coming from e-mail exchanges and some impromptu meetings with a single author was approved by the participants before inclusion to the study data.

## Data analysis

Grounded theory has been indicated to study human behavior (*Easterbrook et al., 2008*), and it is suitable when the research has an explanatory and process-oriented focus (*Eisenhardt, 1989*). Qualitative data analysis techniques from grounded theory responded to our needs (*Langley, 1999*). We are aware that there has been some heated debate regarding which, between *Glaser & Strauss (1967)* or *Corbin & Strauss (2008)*, is *the* grounded theory qualitative strategy (*Creswell, 2009*) or if it can be employed merely as a tool to analyze qualitative data (*Kasurinen, Laine & Smolander, 2013*). *Heath & Cowley (2004)* comparison study concludes that researchers should stop debating about grounded theory, select the method that best suits their cognitive style, and start doing research. We agree with them and adopted *Charmaz (2006)* social constructivist grounded theory approach as a tool to analyze qualitative data coming from face-to-face open-ended interviews, in-field observations, and e-mail exchanges.

The adaption of grounded theory by *Charmaz (2006)* has merged and unified the major coding techniques into four major phases of coding, which are initial coding, focused coding, axial coding, and theoretical coding. The four coding phases have been adopted in the data analysis process of this study. *Charmaz (2006)* has often remembered her readers that no author on grounded theory methodology has ever really offered criteria for establishing what we should accept as a coding family, and that the coding phases are often overlapping, iterative and not strictly sequential within each iteration. This is true also for this study. An exemplar case of our coding activities is shown in Fig. 1. The figure is divided into four columns. The first column provides an interview excerpt. The remaining columns show the intermediate results of the coding activities.

The *initial coding* phase should stick closely to the data instead of interpreting the data. The researchers should try to see the actions in each segment of data, and to avoid applying pre-existing categories to it. Therefore, *Charmaz (2006)* has suggested to code the data on a line-by-line approach so that the context is isolated as much as possible, and to code the data as actions. In order to help focusing on the data as actions, it has been suggested to use gerunds. For example, in Fig. 1 the second column shows the initial codes assigned to a interview snippet.

The second coding phase is the *focused coding*. Focused code means that the most significant or frequent (or both) codes which appeared in the initial coding are employed to sift through larger amounts of data, like paragraphs, speeches, and incidents. This phase is about deciding which initial codes make the most analytic sense for categorizing the data. However, it is also possible to create umbrella codes as substitutes for other codes. During focused coding, the codes become more directed, selective, and conceptual. For example, as shown in Fig. 1, the initial code "Improving productivity through the use of ST" was further abstracted as "Improving productivity through a tool."

The third coding phase is the *axial coding*. The axial coding phase has been proposed by *Strauss & Corbin (1994)*. As synthesized by *Charmaz (2006)*, the axial coding process follows the development of major categories, relates categories to subcategories, and relates them with each others. If during initial and focused coding the data is fractured into pieces,

| Interview snippet | Initial coding | Focused coding | Axial coding |
|---|---|---|---|
| [Interviewer: "Do you think that Sublime Text is enhancing your productivity then?"] | | | |
| P2: "Absolutely. I was extremely excited by these features and they pushed me to do more and more." | Improving productivity through the use of ST; being motivated by ST to do more work; | Improving productivity through a tool; Feeling gratitude towards a tool; feeling motivated because of a tool | PERFORMANCE_positive; EVENT_using_useful_tool; AFFECT_gratitude; AFFECT_motivated; |
| [Interviewer: "Were you actually thinking about this while you were working?"] | | | |
| P2: "Definitely. First, I turned the monitor towards P1 and showed him the magic. But I felt good for the rest of the day, and I accomplished more than what I hoped I could do." | Thinking about the improved performance brought by a tool; showing the features of a tool to a team mate; Feeling good during a workday because of tool functionality; accomplishing more than what planned; | Realizing positive performance; Sharing information; Feeling strongly good; Progressing strongly on goal; | ATTRACTOR_good; PERFORMANCE_positive; FOCUS_positive; GOAL_progressing; |

**Figure 1** **Example of coding phases for this study.**

the axial coding phase brings the data back together again. In this phase, the properties and the dimensions of a category are specified. The fourth column of Fig. 1 shows an iteration of axial coding.

The fourth coding phase is the *theoretical coding*. Theoretical coding was introduced by *Glaser (1978)*. As synthesized by *Charmaz (2006)*, the theoretical coding phase specifies how the codes from the previous phases related to each other as hypotheses to be integrated into a theory.

It would be impractical to show the steps and complete examples of axial and theoretical coding as they would need several interview excerpts and resulting codes (*Charmaz, 2006*). What we could demonstrate in Fig. 1 was that the interview excerpt was further coded in the later coding phases and became part of the evidence to support the key concepts, such as affect, and their components as shown in the fourth column. The overlapping of different categories over the same snippets indicated the potential linkage among them, which became the basis to develop the model proposed in this study.

## Reliability

Here, we describe our procedures for enhancing the reliability of the gathered data and the results. The data was gathered using multiple sources. Each interview was accompanied by handwritten notes, recordings, and related subsequent transcriptions. All in-field observations were accompanied by audio recordings after obtaining permission of the

participants. We wrote memos during the study. The transcriptions and the coding phases were conducted using *Atlas.ti 7.5*, which is a recognized instrument for such tasks.

In order to make the participants focus on their affects and recall how they felt during performance episodes, we asked them to fill out a very short questionnaire at each git commit. The questionnaire was the Self-Assessment Manikin (*Bradley & Lang, 1994*), which is a validated pictorial questionnaire to assess affects. We employed the questionnaire in a previous study (*Graziotin, Wang & Abrahamsson, 2015a*) as it proved to be quick (three mouse clicks for completing one) and not invasive. We employed the gathered data to triangulate the observational data and the interview data during each interview. If there was disagreement between the qualitative data (e.g., several positive affective episodes but negative quantitative results), we asked for further clarification from the participants to solve the discrepancies.

As a further action to enhance reliability, but also ethicality of the study, we asked the participants to individually review the present paper in three different times. The first review session happened in the initial drafts of the paper when we solely laid down the results of the study. The second review session happened right before submitting the article. The third review session happened before submitting a revised version of the present article. For the reviews, we asked the participants to evaluate the results in terms of their own understanding of the phenomena under study and the protection of their identity and dignity. Because of their valuable help, the proposed theory is shared with them and further validated by them.

## RESULTS AND DISCUSSION

The study was set in the context of a Web- and mobile-based health-care information systems development between July and September 2014. Two software developers, who were conducting a semester-long real-world project as a requirement for their BSc theses in Computer Science, were put in a company-like environment. Both developers, who we shall call P1 and P2 for anonymity reasons, were male. P1 was 22 years old and P2 was 26 years old. They both had about five years of experience developing Web and mobile systems. P1 and P2 had their own spacious office serving as an open space, their own desks and monitors, a fast Internet connection, flip-charts, a fridge, vending machines, and 24/7 access to the building. The developers accepted to work full time on the project as their sole activity. They were instructed to act as if they were in their own software company. Indeed, the developers were exposed to real-world customers and settings. The customers were the head of a hospital department, a nurse responsible for the project, and the entire nursing department. The development cycle began with a first meeting with the customer, and it ended with the delivery of a featureful first version of the working software.

It is beneficial to the reader to provide a brief summary of the main events, which have been extracted from our in-field memos. During the first week, P1 had to work on the project without P2. P2 failed to show up at work. During the first days, P2 gave brief explanations about the absence, e.g., housework or sickness. However, the explanations stopped quickly, and P2 stopped answering to text messages and phone calls. At the

beginning of the second week, P2 showed up at work. P2 had some private issues, which brought some existential crisis. P1 was initially reluctant to welcome P2 in the development, as all the code so far was P1's creation. The first two days of collaboration brought some tension between the team members, crippled experimentation with the code, and a shared loss of project vision. On the third day of the second week, the team tensions exploded in a verbal fight regarding the data structures to be adopted. At that point, one of the present authors was involved in the discussion. The researcher invited the participants to express their opinion and acted as mediator. A decision was eventually made. The initial tensions between the developers began to vanish, and the work resumed at a fair pace. At the end of the second week, P1 and P2 had a further requirements elicitation session with the customer represented by the head nurse. The development appeared to be back at full speed, and a full reconciliation could be observed between the participants. The progresses succeeded one day after another, and the fully working prototype was demoed and tested during the sixth week.

Face-to-face open-ended interviews happened at the beginning of the project during 11 scheduled meetings and 5 impromptu shorter meetings called by the researchers or by the participants. The impromptu meetings were held mostly because of trivial issues, like casual chatting which turned into a proper interview. Only in one case was an impromptu meeting called by P2 when he finally came back to work. We also did not distinguish between the data coming from the scheduled meetings and the impromptu meetings. The interviews were open-ended and unstructured, but they all began with the question *How do you feel?* In-field observations happened on an almost daily basis. The participants were informed if they were recorded. We recorded a total of 657 min of interviews. Finally, data was gathered via the exchange of thirteen emails.

The transcripts of the interviews were completed immediately after the interviews were concluded. The initial coding phase produced 917 unique codes. The focused coding phase was focused on the individual's experiences of the development process, and it produced 308 codes. Figure 1 provides an example of our coding activities. The axial coding and theoretical coding produced six themes, which are explained in this section. Inconsistencies between the qualitative data and the data from the Self-Assessment Manikin questionnaire happened three times during the entire study. All three discrepancies were minor, and they were immediately solved upon clarification from the participants. For example, in one case the participant P1 reported low values of valence and arousal, and a neutral value for dominance. During the interview, P1 often stated that he had a frustrating day, but there were no mentions of low-arousal negative affects. When asked to explain how the Self-Assessment Manikin values were representative of the work day, the participant added that he felt low esteem, which was caused by episodes of frustration. Overall, P1 was unexcited and lost over the day; thus the reported low value for arousal.

This section provides the proposed theory. The theory is represented in Fig. 2. We describe the discovered themes and categories (boxes) and their relationships (arrows). While Type II theories are not expected to discuss causal explanations in terms of direction and magnitude (*Gregor, 2006*), we offer them as they were interpreted from the data. Each

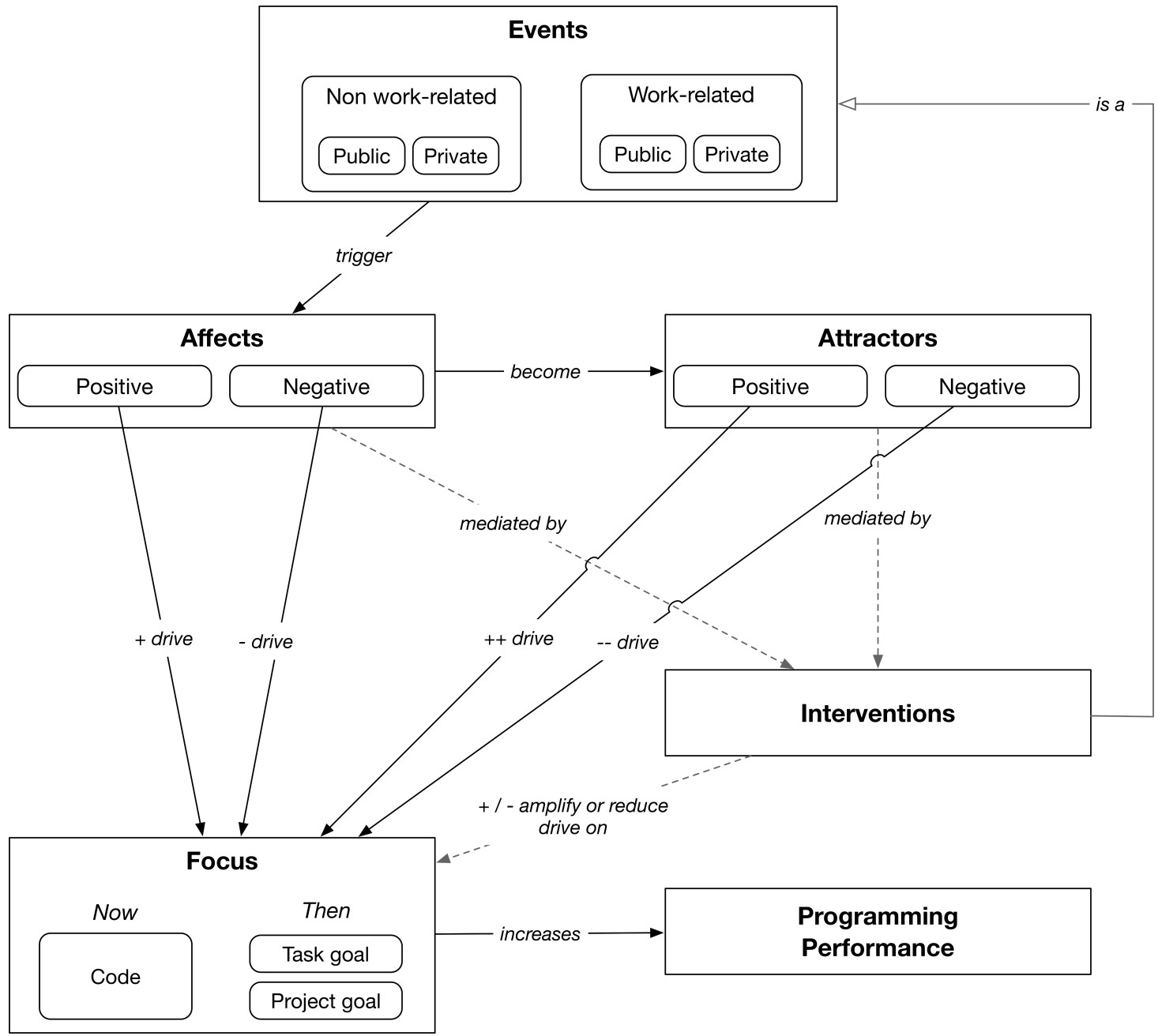

**Figure 2** A theory of the impact of the affects on programming performance.

relationship is accompanied by a verb, which describes the nature of the relationship. Where possible, we precede the verb with some plus (+) or minus (−) signs. A plus (minus) sign indicates that we theorize a positive (negative) effect of one construct to another. A double plus (double minus) sign indicates that we theorize a strong positive (strong negative) effect of one construct to another with respect to a proposed weaker alternative. The reader should bear in mind that our theorized effects are not to be strongly

interpreted quantitatively. That is, a double plus sign is not the double of a single plus sign or an order more of magnitude of a single plus sign. Every entity and relationship is supplied with interview quotes, codes, and related work.

## Events

The *events* are perceived from the developer's point of view as something happening. Events resemble *psychological objects*, which were defined by *Russell (2003)* as "the person, condition, thing, or event at which a mental state is directed" (p. 3) but also at which a mental state is attributed or misattributed.

Events may be *non work-related*—e.g., family, friends, house, hobbies—or they may be *work-related*—e.g., the working environment, the tools, and the team members. The interview quotes 1 and 2, and in-field memo 3 are examples of work-related events, while interview quote 4 is not related to work.

1. "*Suddenly, I discovered Google Plus Bootstrap, which is a Bootstrap theme resembling Google+. [I implemented it and] it was easy and looking good.*"—P1
2. "*I found a typo in the name of the key which keeps track of the nurse ID. The bug was preventing a correct visualization of patient-related measurements. Fixing the bug is very satisfying, because I can now see more results on the screen.*"—P2
3. P1, talking to P2 and visibly irritated "Again this? You still have not understood the concept! It is <component name> that is static, while the measurement changes!"
4. "*This morning I received a message with some bad news related to my mother. I immediately desired to abandon development in order to solve the possible issue. The focus was more on that issue than on any other issue at work.*"—P1

We further distinguish public events from private events. *Public events* are those that could be observed by a third person. The in-field memo 3 is an exemplar public event. *Private events* are known to oneself only, even if they are coming from the real world. For example, the event described in interview quote 4 was real and coming from the real world. However, it was not observable by a third person. Events have often an episodic nature, as P1 and P2 noted on several occasions. However, private events can also be reflections, realizations, memories, and situations as with psychological objects.

5. Interviewer: "*Have you focused better on your programming task today?*" P2: "*Yes, today went better [than usual]. It's probably when you do that [programming] alone that I am more.. it is more difficult, to write code. When I am working with somebody it goes better, you can work better.*"

In the interview quote 5, P2 described the general situation, or a summary of the work day events with respect to usual situations. Situations can be causation chains or aggregation of previous events. The participants do not need to be aware of events as merely events or as situations as it does not make any difference to them. We are not representing situations in Fig. 2 because we still consider them as events. The rest of the paper provides numerous other examples of events.

## Affects

During the development process, several *affects* have been triggered by events and felt by the developers. We coded only affects which had been directly mentioned by P1 and P2.

The following are the detected positive and negative affects (respectively) being felt during the development cycle.

*accompanied, accomplished, attracted, contented, dominating, enjoyed, excited, fun, good, gratitude, happy, illuminated, motivated,[5], optimistic, positive, satisfied, serene, stimulated, supported, teased, welcomed.*

*angry, anxious, bored, demoralized, demotivated, depressed, devastated, disinterested, dominated, frustrated, guilty, loneliness, lost, negative, pissed off, sad, stagnated, unexcited, unhappy, unsatisfied, unstimulated, unsupported, worried.*

Our qualitative results on the perceived affects agree with the quantitative results of *Wrobel (2013)* and *Müller & Fritz (2015)*, which indicated that developers do feel a very broad range of affects in the software development process.

Examples of events that caused positive and negative affects (respectively), coded using the gerund principle of *Charmaz (2006)* method for analyzing qualitative data, are the following.

*'Feeling contented because a very low number of code changes caused big achievement in terms of quality [or functionality],' 'Feeling gratitude towards a tool,' 'Feeling attracted by a junk of code because of anticipating its value for the end user,' 'Feeling motivated because personal issues are now out clear,' 'Feeling supported because of the brought automation of a framework,' 'Feeling serene because of a low workload right after a high workload,' 'Feeling happy because of sensing the presence of a team member after reconciliation.'*

*'Feeling alone [or unsupported] while working [or by a team member],' 'Feeling anxious because of a sudden, not localizable bug that ruined the day,' 'Feeling anxious by not understanding the code behavior,' 'Feeling bored by implementing necessary but too static details [e.g., aesthetic changes instead of functionalities],' 'Feeling frustrated by the different coding style of a team member,' 'Feeling angry by failing to integrate [or extend] an external component,' 'Feeling stagnated in life [or job, or studies],' 'Feeling unstimulated because of a too analytic task.'*

According to previous research, psychological objects—sometimes in the form of events, sometimes as stimula—trigger affects all the time, and an individual is under a particular affect or a blend of affects all the time (*Russell, 2003*). Sometimes, these affects will be perceived strongly. Sometimes, they will not be perceived at all despite their presence. A failure to attribute an affect to an event does not demise the affect itself. This affect misattribution coincides with some theories of moods (*Fisher, 2000*; *Weiss & Cropanzano, 1996*), which consider affect as non attributed emotions or simply as free-floating, unattributed affect (*Russell, 2003*).

[5] The careful readers might turn up their nose here. As we wrote in *Graziotin, Wang & Abrahamsson (2015b)* affects are not motivation, as they are not job satisfaction, etc. Yet, affects are important components of these psychological constructs, and studying complex multifaceted constructs like motivation would require different approaches and different measurement instruments. For this reason, if the participants only stated that they felt motivated or satisfied, we considered them as affects, as it might well be the case that they were expressing emotional judgments about such constructs. In any case, the inclusion or exclusion of such terms as affects would not change the results of this study.

## Attractors

We observed that some events had a particular affective meaning to the participants. These affective experiences were assumed high importance to the participants with respect to other affective experiences; thus, we called them *attractors*.

Attractors are affects, which earn importance and priority to a developer's cognitive system. At a very basic instance, they gain the highest possible priority and emphasis to a developer's consciousness, to the point that behaviors associated to the attractor can be observed as it is experienced. An example can be offered by quote 6 below.

6. P2: "*I did a really good job and fixed things also due to Sublime Text (ST).*" Interviewer: "*What has ST done for you?*" P2: "*When you copy/paste code around and refactor, ST offers you at least three different ways for doing search and replace. It is really advanced.*" Interviewer: "*Would another tool make a difference to your work instead?*" P2: "*With another editor or an IDE it would be another story, especially if an editor tries to do too much, like Eclipse. I think that the compromise between functionality and usability of ST is way better.*" Interviewer: "*Do you think that ST is enhancing your productivity then?*" P2: "*Absolutely. I was extremely excited by these features and they pushed me to do more and more.*" Interviewer: "*Were you actually thinking about this while you were working?*" P2: "*Definitely. First, I turned the monitor towards P1 and showed him the magic. But I felt good for the rest of the day, and I accomplished more than what I hoped I could do.*"

In interview quote 6, P2 offered an insight regarding the affects triggered by a software development tool. The excitement toward the tool features was an attractor to P2. The attractor became central to the developer subjective conscious experience, not just an underlying affect. Moreover, the behavior caused by the experience of the attractor was directly observable. Interview quote 6 emphasizes that attractors are not necessarily concerns or negative in nature.

Interview quote 4 provides instead an example of a negative attractor. P1 realized that a non work-related event was not desirable, thus generating negative affects. What happened to his mother was important and demanded his attention. P1 was consciously experiencing the negative attractor, and the appraisal of such attractor had consequences to his way of working.

Attractors are not necessarily stronger than general affects for gaining a developer's subjective *conscious* experience. They might just *be there* and still have an impact. We can access them retrospectively. Interview quote 7 is an example of such occurrence.

7. "*I am not progressing.. in the working environment.. with my university career. With life. I feel behind everybody else and I do not progress. And I am not even sure about what I want to do with my life. I got no visual of this.*"—P2

Moreover, interview quote 7 shows that attractors are not always caused by single events. Attractors can become reflections on a series of events as a consequence of them and as a summation of them.

Another example of reflections of a series of events that have however an impact on a developer's subjective consciousness is shown in interview quote 8. P2 was having a life crisis which resulted in a loss of the vision of his own life.

8. "*When I was alone at home, I could not focus on my programming task. The thought of me not progressing with life did often come to my mind. There I realized that I was feeling depressed.*"—P2

In interview quote 8, the participant had a negative *depressed* attractor with the attached meaning *I am not progressing with life*. The rumination associated with this attractor was strong and pervaded P2 personal experience and his everyday life of that period.

Attractors are part of the personal sphere as much as affects are—indeed, they are special affects for us. In the software process improvement literature, the term *concern* has been used as commitment enabler (*Abrahamsson, 2001*). The commitments are formed in order to satisfy such concerns, i.e., needs (*Flores, 1998*). Attractors are not concerns as employed by *Abrahamsson (2001)*. An important difference is that concerns are linked to actions, i.e., actions are driven by concerns. On the other hand, attractors are affects, and affects are not necessarily concerns, nor do they necessarily cause immediate actions.

Under our current theoretical framework, a blend of affects constitutes an individual's happiness, at least under the hedonistic view of happiness (*Haybron, 2001*). According to this view, being happy coincides with the frequent experience of pleasure; that is, happiness is reduced to a sequence of experiential episodes (*Haybron, 2001*). Frequent positive episodes lead to feeling frequent positive affects, and frequent positive affects lead to a positive *affect balance* (*Diener et al., 2009*). *Lyubomirsky, King & Diener (2005)* consider a person *happy* if the person's affect balance is mainly positive. However, we have just stated in this section that some developers' affects are more important than other affects. Let us now be more specific.

As argued by the philosopher *Haybron (2001)*, a quantitative view of happiness based solely on frequency of affects is psychologically superficial because some affects do not have distinct episodes or attributions (as in moods). Even more, *Haybron (2005)* has seen happiness as a matter of a person's affective condition where only *central affects* are concerned. We see a similarity between attractors and *Haybron (2005)* central affects. As attractors are important affects, we agree that they are a strong constituent of the happiness of the individuals. However, non attractors could be central affects, as well. In our observations, we saw that attractors are also affects that are easily externalized by the participants, and we will show that their originating events are more visible to them. Furthermore, we will show that attractors are more linked to the focus and the developers' performance. Thus, we differentiate them from central affects.

The participants could sometimes realize the affective meaning of attractors by themselves, as in quote 8. There is often the need to externalize them in order for an observer to feel them. We found that sometimes, externalizing affects is alone beneficial, as seen in the next section.

## Interventions

While the presence of researchers has always an influence on the participant's behaviors (*Franke & Kaul, 1978*), it happened twice that our interaction with the participants had a clear effect on their feelings and behaviors. We call such events *interventions*. Interventions are events—as shown in Fig. 2 by the UML-like grey arrow with a white arrowhead—that mediate the intensity of already existing negative attractors, thus reducing them as much as possible to normal affects. After externalizing his depressed state in interview quote 8, P2 continued as follows:

9. "*What we were doing was not 'in focus.' The result really didn't matter to me. To my eyes, we were losing time. However, once I've told you what I told you [the personal issues] you know that as well. It is not that I am hiding or that I am inventing things out..I now have no more the possibility to wriggle anymore. I told you why I was not there and I am feeling better already. I am now here for two days, and I feel way better than before.*"—P2.

The field memos provided more evidence on the effectiveness of interventions. For example, during the reconciliation, which happened at the beginning of week 2, the developers had frequent soft fights.

P2 battles fiercely for his opinions and design strategies. However, he is listening to P1 opinions. On the other hand, P1 seems more interested to get stuff done, and he seems less prone to listen to P2. P2 is probably realizing this and responds using passive-aggressive modes. Some not-so-very nice words fly.

P1 and P2 are less aggressive with each other. My proposal to let them express their opinions and to invite them to listen to each other seems to have a positive effect. A solution, albeit influenced by me, seems to have been reached.

A field memo six days after the reconciliation was much more positive.

P1 and P2 have been working with an almost stable pace. There does not seem to be an elephant in the room anymore. Both of them smile often and joke with each other. You can feel them happier than before. I often see P1 and P2 showing their results to each other. The work seems way more productive than last week.

Even personal issues were having less impact on P2 as he revealed in an interview nine days after the reconciliation.

10. "*My personal issues are having a minor impact on my productivity, despite the fact that my mind wonders in different places. It is because we are now working well together and share a vision.*"—P2

Interventions in Fig. 2 are reached by dashed arrows, which start from affects and attractors, and have a dashed arrow pointing to focus. The dashed arrows, together with the labels *mediated by* and *amplify (or reduce) drive on*, indicate alternative paths in the process. That is, affects and attractors are mediated by interventions, which amplify or reduce their drive on the focus.

These interventions suggest that a mediator is a useful figure in a software development team. The mediator should be able to gently push the team member to let out their opinions, views, and affects. A more concrete example could be an agile coach or a team leader according to the team settings.

## Focus—progressing and goal setting

In this section, we explain the construct of focus, which is related to progressing toward goals and the setting of such goals. The *focus* often referred to a general mental focus, e.g., "*I was in focus after I could refactor all that code using Sublime Text search-and-replace capacity.*"—P2, which usually matched a focus on the current chunk of code. However, the focus on the current chunk of code was with respect to a goal. P2 mentioned focus in interview quote 8, where he told the interviewer that he could not focus on the programming task while at home, because of the realization of being depressed. A more tangible focus on the code at hand was portrayed by P1 in the following interview quote.

11. "*After our [between P1 and P2] reconciliation and after the meeting with [the head nurse], I often developed in full immersion. When I am in full immersion mode, nothing exists except what I am doing. I have a goal in mind and I work toward it. I don't think about anything else but my goal and my progress towards it.*"—P1

During the last interview, P1 was directly asked about the way he focuses while developing software and what he thinks about. Besides the full immersion mode that P1 described in quote 11, he described a "*lighter mode of immersion. I enter this mode when I am tired, when I write less functional aspects of the code.*" but also "*when I am interrupted by negative news or when I focus my attention more on some problems.*"

In quote 12, P2 shared his view on negative affects and how they hinder performance by changing the way he perceived events as attractors.

12. "*My negative thoughts have been the same lately—more or less–but I sometimes change the way I look at them. It is often positive, but it is often negative, too. Maybe I realize this more when I have a negative attitude towards them. It influences my work in a particular way: my concerns become quicksand.*"—P2

Our *focus* appears to be similar to the flow as depicted by *Csikszentmihalyi (1997)*, and found in the related work by *Meyer et al. (2014)* and *Müller & Fritz (2015)*, which was described as an attention state of progressing and concentration.

Additionally, the participants often mentioned the term 'vision,' which was meant as the "*ability to conceive what might be attempted or achieved.*" (*OED Online, 2015*). For this reason, we preferred using the term *goal setting*. The participants linked the focus and the capacity of setting goals. Goal settings has an established line of research in organizational behavior and psychology—one of the seminal works is by *Locke (1968)*—that would deserve its own space in a separate article. It involves the development of a plan, which in our case is internalized, designed to guide an individual toward a goal (*Clutterbuck, 2010*). Those goals found in our study were related to future achievements in the short and

long run, i.e., the task and the project. One example of task goals lies in the interview quote 13. Whenever the focus of attention was on the current code melted with the goal setting of task and project, the performance was reported and observed as positive. However, if something was preventing the focus on the current code—*now*—and the focus on the goal or the goal setting of the task or project—*then*—the performance was reported and observed as negative. P2 summarized these reflections concisely in quote 13.

13. "*It does not matter how much it is actually going well with the code, or how I actually start being focused. Then it [my thoughts about my personal issues] comes back into mind. It is like a mood. I cannot define it in any way. But it is this getting rid of a thought, focusing back to work and the task goal. Here [shows commit message] I wanted to add the deletion of messages in the nurses' log. But when it happens, I lose the task vision. What was I trying to accomplish? WHY was I trying to do this? It happens with the project vision, too. I don't know what I am doing anymore.*"—P2

The project goal setting is similar to the task goal setting. The difference is that project goal setting is the capacity of perceiving the completion of a project in the future and visualizing the final product before its existence as P1 outlined in interview quote 14.

14. "*After we talked to [the head nurse], we gathered so much information that we overlooked or just did not think about. [...] between that and the time you [the researcher] invited us to speak about our issues and mediated among our opinions, we had a new way to see how the project looked like. The product was not there still, but we could see it. It was how the final goal looked like.*"—P1

There is a link between focusing on the code and focusing on the task goal. Staying focused on the code meant staying focused on the *now* (and here). It is the awareness of the meaning of each written line of code towards the completion of a task. Focusing on the task and project goals meant staying focused on the *then* (and there). It was meant as the capacity of envisioning the goal at the shorter term (the task) and the overall goal of the project. At the same time, focusing on the task and the project meant the possibility to define a task completion criteria, the awareness of the distance towards the completion of such task, and to re-define the goal during the work day.

Our findings are in line with those of *Meyer et al. (2014)*, where the participants in a survey perceived a productive day as a day where "they complete their tasks, achieve a planned goals or make progress on their goals" (p. 21). The number of closed work items, e.g., tasks and bugs, was the most valued productivity measurement among developers. The *full immersion mode* mentioned by P1 in interview quote 11 resembles the flow depicted by *Csikszentmihalyi (1997)* and mentioned in the related work by *Meyer et al. (2014)* and *Müller & Fritz (2015)*.

## Performance

The performance was generally understood by the participants as their perceived effectiveness in reaching a previously set expectation or goal. Or, whenever *then* became *now*.

15. "*Last week has been chaotic. We worked very little on the code. P2 played around with the programming framework. P2 tried to adapt an example program to fit our needs. So, P2 studied the chosen framework. I can say that P2 was productive. I spent my time doing refactoring and little enhancements of what was already there. Little functionality was developed so far. In a sense, we still performed well. We did what we were expecting to do. Even if I did so little. I still laid down the basis for working on future aspects. So yeah, I am satisfied.*"—P1

16. Interviewer: "*What happened during this week?*" P2: "*Well, it happened that..I did not behave correctly in this week. I could not do a single commit.*"

We observed that the affects have an impact on the programming performance of the developers. This is achieved by driving the focus that developers have on the currently focused code, the ongoing task, or the project itself.[6] P2 suggested already, in interview quote 6, that the excitement caused by the discovery of the useful search-and-replace functionalities in his editor had pervaded his work day. This positive attractor caused him to be productive also when not using such functionalities. P2 could also offer cases of the opposite side, like the one in quote 17.

17. "*I was lost in my own issues. My desire to do stuff was vanishing because I felt very depressed. There was not point in what I was currently doing, to the point that I could not realize what I had to do.*"—P2

More precisely, positive affects have a positive impact on the programming performance—as they drive the focus positively—while negative affects have a negative impact on the programming performance—as they drive the focus negatively. While most of the previous quotes are examples on the negative side, quote 6 and the following quote are instances of the positive case.

18. P1: "*I now feel supported and accompanied by P2. We are a proper team.*" Interviewer: "*What has changed?*" P1: "*It's that now P2 is active in the project. Before [the reconciliation] P2 was not here at all. [ . . . ] If he joined after our meeting with [the head nurse], there was the risk to see him as an impediment instead of a valid resource and team member. Now, I feel happier and more satisfied. We are working very well together and I am actually more focused and productive.*"

A positive focus has a positive effect on programming performance. But, a focus on the code toward a task or project goals (or a combination of them) have an even stronger positive impact on the programming performance.

We provide some codes related to the consequences of positive and negative affects (respectively) while programming.

'*Limiting the switch to personal issues because of feeling accompanied by a team member,*' '*Switching focus between the task and the positive feelings caused by a tool makes productive,*' '*Focusing better on code because of the positive feelings brought by reconciliation,*' '*Focusing*

[6] The aim of this study is to offer a theory of the impact of affects on performance while programming rather than proposing a performance or productivity theory. A plethora of factors influence the performance of developers—see *Wagner & Ruhe (2008)*; *Sampaio et al. (2010)* for a comprehensive review of the factors—and affects are one of them, although they are not yet part of any review paper. At the same time, software development performance is composed of several complex interrelated constructs—see *Petersen (2011)* for a review of productivity measurements—to which we add those driven by cognitive processes and *also* influenced by affects, e.g., creativity and analytic problem solving capability (*Graziotin, Wang & Abrahamsson, 2014a*).

*less on personal issues [more on the code] because of a sense of being wanted at work,' 'Focusing more on code because of feeling supported and in company,' 'Committing code frequently if feeling in company of people.'*

*'Abandoning work because of negative feelings fostered by negative events,' 'Avoiding coming to work because of lost vision [and depression],' 'Avoiding committing working code during day because of loneliness,' 'Choosing an own path because of the loneliness,' 'Switching focus between personal issues and work-related task prevents solving programming tasks,' 'Losing focus often when feeling alone,' 'Losing the project vision because of quicksanding in negative affects,' 'Not reacting to team member input because of bad mood,' 'Realizing the impediments brought by personal issues when they are the focus of attention,' 'Trying to self-regulate affects related to negative events and thoughts lowers performance,' 'Underestimating an achievement because of loneliness,' 'Worrying continuously about life achievements and avoiding work.'*

### Comparison of the theory with related work

The proposed theory can be seen as a specialized version of Affective Events Theory (AET, (*Weiss & Cropanzano, 1996*)). It provides an affect-driven theory explaining how events, both work-related and not, impact the performance of developers through their focus and goal setting while programming. Therefore, our study produces evidence that AET is an effective macrostructure to guide research of affects on the job in the context of software development. At the same time, our proposed theory is reinforced by the existence of AET itself.

We also note that our theory is partially supported in *Müller & Fritz (2015)* independent study—built upon one of our previous studies (*Graziotin, Wang & Abrahamsson, 2015a*)—which was conducted at about the same time of the present study.[7] Among their findings, the self-assessed progressing with the task is correlated with the affects of developers; the most negative affects were correlated with less focus on clear goal settings and positive affects were linked with focusing and progressing toward the set goals.

Finally, our findings are in line with the general findings of goal settings research. That is, the task performance is positively influenced by shared, non conflicting goals, provided that there are fair individuals' skills (*Locke & Latham, 2006*).

### Happy, therefore productive or productive, therefore happy?

Let us now reason a little on the causality aspects between affects and performance. We note that the participants have always explicitly stated or suggested that the influence of affects on performance is of a causality type. Some researchers have warned us that there might instead be a correlation between the constructs, as well as a double causality (*I am more productive because I am more happy, and I am more happy because I am more productive*). Indeed, so far in our previous studies (*Graziotin, Wang & Abrahamsson, 2014a*; *Graziotin, Wang & Abrahamsson, 2015a*) we have argued for correlation, not causation.

In the present study, we could not find support in the data for a double causation, but for a causality chain *Happy, therefore productive*, in line also with related research

7 Furthermore, at our submission time the work by *Müller & Fritz (2015)* had just been publicly accepted for inclusion in ICSE 2015 proceedings, but it is currently still not published formally. We obtained their work through an institutional repository of preprints.

(*Wrobel, 2013*). However, it seems reasonable that we are happier if we realize our positive performance.

We speculate here that a third, mediating option might exist. In the proposed theory, and in several other theories in psychology, being happy is reduced to frequent feeling of positive affects (*Haybron, 2001*). As argued by *Haybron (2007)*, the centrality of affects might be relevant, as well. *Haybron (2007)* stated, as an example, that the pleasure of eating a cracker is not enduring and probably not affecting happiness; therefore, it is considered a peripheral affect. Peripheral affects arguably have smaller—if not unnoticeable—effects on cognitive activities. It might be the case that the positive (negative) affects triggered by being productive (unproductive) do exist but have a small to unnoticeable effect on future productivity. However, this is outside the scope of this study. We report our backed up speculation as causation for future work.

## CONCLUSION

In this qualitative, interpretive study, we constructed a theory of the impact of affects on software developers with respect to their programming performance. As far as we know, this is the first study to observe and theorize a development process from the point of view of the affects of software developers. By echoing a call for theory building studies in software engineering, we offer first building blocks on the affects of software developers. For this reason, we designed our theory development study using a small sample adhering to guidelines for generating novel theories, thus enabling the development of an in-depth understanding of an otherwise too complex and complicated set of constructs.

The theory conceptualization portraits how the entities of events, attractors, affects, focus, goal settings, and performance interact with each other. In particular, we theorized a causal chain between the events and the programming performance, through affects or attractors.

Positive affects (negative affects) have a positive (negative) impact on the programming task performance by acting on the focus on code, and task and project goals. The theory introduces the concept of attractors, which are affects that earn importance and priority to a developer's cognitive system and, often, to their conscious experience. Attractors have an even higher impact on programming performance than ordinary affects.

Finally, we also provided evidence that fostering positive affects among developers boosts their performance and that the role of a mediator bringing reconciliations among the team members might be necessary for successful projects.

### Contributions and implications

Our study offers multiple contributions and implications. The theoretical contributions lie in the theory itself. The theory incorporates the impact of affects on performance through an influence on the focus of developer's consciousness on coding and on several aspects of goal settings (task, project). In addition, we introduced the concept of attractors for developers, which is a novel construct based on affects and events at different spheres (work-related and not, private or public). The theory is proposed as part of basic science of software engineering, and it is open to falsification and extension.

As stated by Lewin, "there is nothing quite so practical as a good theory" (*Lewin, 1945*). The practical implication of our study is that, despite the idea among managers that pressure and some negative feelings help in getting the best results out, there is growing evidence that fostering (hindering) positive (negative) affects of software developers has a positive effect on the focus on code, and task and project goal settings, and, consequently, on their performance. Additionally, we found evidence that a mediator role to reconcile the developers' issues and conflicts is a way to foster positive affects and mediate negative attractors among them.

The proposed theory can be employed as a guideline to understand the affective dynamics in a software development process. The theory can be used to foster a better environment in a software development team and to guide managers and team leaders to enrich their performance by making the developers feel better. On the other hand, our conceptualized theory can guide the team leaders to understand the dynamics of negative performance when it is linked to negative affects.

## Limitations

The most significant limitation of this research to be mentioned lies in its sample. Although it is very common for software engineering studies to recruit computer science students as participants to studies (*Salman, Misirli & Juristo, 2015*), for some readers this might still be considered a limitation. First, it is true that our participants were enrolled to a BSc study in computer science, but they both had a working history as freelancers in companies developing websites and Web applications. While our developers did not have to be concerned about assets and salaries, they were paid in credit points and a final award in terms of a BSc thesis project. *Tichy (2000)* and *Kitchenham et al. (2002)* argued that students are the next generation of software professionals as they are close to the interested population of workers, if not even more updated on new technologies. Indeed, the empirical studies comparing students in working settings with professionals did not find evidence for a difference between the groups (*Svahnberg, Aurum & Wohlin, 2008*; *Berander, 2004*; *Runeson, 2003*; *Höst, Regnell & Wohlin, 2000*; *Salman, Misirli & Juristo, 2015*). The conclusions from the previous studies are that students are indeed representatives of professionals in software engineering studies.

The non-inclusion of female participants might be considered a further limitation of this study. There is a widespread popular conception that there are gender differences in emotionality (*McRae et al., 2008*). Evidence has been found for gender differences at the neural level associated to reappraisal, emotional responding and reward processing (*McRae et al., 2008*), and for a female having greater reactivity to negative stimuli (*Gardener et al., 2013*) and adoption of different emotion regulation strategies (*Nolen-Hoeksema & Aldao, 2011*). While more studies on gender differences are needed as the produced evidence is not enough yet (*Nolen-Hoeksema, 2012*), it might be the case that the inclusion of a female developer would have made the dataset richer, and perhaps would have led to a more gender-balanced theory.

While we argued extensively about the choice of the sample size in section *Theory Construction and Representation*, we repeat here that there was the need to keep the phenomenon under study as simple as possible given its complex nature (*Barsade & Gibson, 1998*). Furthermore, when novel theories are to be developed in new domains, such as software engineering, a small sample should be considered (*Järvinen, 2012*). This strategy, while sometimes seen as limiting, pays off especially for setting out basic building blocks (*Weick, 1995*). As argued by *Bendassolli (2013)*, even one observation could be sufficient for theorizing as so far as "phenomena should be directly explained by theory, and only indirectly supported by the data" (quoted from Section 6.2). Our choice of the small sample size was seen as a benefit for the purposes of this explanatory investigation. The reason is that in a real company, the source of events is vast and complex. There are team dynamics with complicated, and even historical, reasons that are harder to grasp—triggering a complex, powerful network of affects (*Barsade & Gibson, 1998*)—thus lifting the study's focus out from the programming itself.

## Future work

We have three directions of research to suggest to the readers. The first one is an immediate continuation of our study. As our study was explanatory, we suggest future research to test the proposed theory and to quantify the relationships in quantitative studies, in software engineering field but also in other domains to understand if and how the specifics particular to the software engineering context affect the applicability of our theory. Although quantifying the impact of attractors was beyond the scope of this study, we feel that negative attractors triggered by non work-related events and positive attractors triggered by work-related events have the strongest impact on the performance of software developers. Furthermore, this study focused on the dimensions of positive and negative affects. It is expected that different types of affects and attractors matter more than other, and have different impact on the focus and performance. We leave future studies the option to study discrete affects, e.g., joy, anger, fear, frustration, or different affect dimensions, e.g., valence, arousal, and dominance.

Our second suggestion for future studies is to focus on dynamic, episodic process models of affects and performance where time is taken into consideration. The underlying affects of developers change rapidly during a workday. The constituents and the effects of such changes should be explored. Additionally, exploring the dynamics of affects turning into attractors (and possibly vice-versa) and what causes such changes will provide a further understanding of the effectiveness of interventions and making developers feeling happier, thus more productive.

Finally, our third direction for future research is to broaden the focus on (1) artifacts different than code, such as requirements and design artifacts, and (2) understanding the complex relationship of affects and software developers' motivation, commitment, job satisfaction, and well-being.

## ACKNOWLEDGEMENTS

We thank our two participants, who openly, actively, and unhesitatingly collaborated during the research activities. We are grateful for the feedback provided by two anonymous reviewers, which let us improve the manuscript in terms of several aspects including clarity.

### Funding

The authors received no funding for this work.

### Competing Interests

The authors declare there are no competing interests.

### Author Contributions

- Daniel Graziotin conceived and designed the experiments, performed the experiments, analyzed the data, wrote the paper, prepared figures and/or tables, performed the computation work, reviewed drafts of the paper.
- Xiaofeng Wang performed the experiments, analyzed the data, wrote the paper, reviewed drafts of the paper.
- Pekka Abrahamsson analyzed the data, wrote the paper, reviewed drafts of the paper.

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
