# Peer review of "How do you feel, developer? An explanatory theory of the impact of affects on programming performance"

_PeerJ Computer Science, doi:10.7717/peerj-cs.18_

## Round 0.1 · original submission · Minor Revisions

Thank you for this excellent contribution to PeerJ Computer Science. As you can see below, both anonymous reviewers have enjoyed the paper a lot. However, both have requested minor updates prior to publication.

Specifically, I ask you to address the comment by Reviewer 1 relating to the transparency of some of the interpretation steps. I agree with Reviewer 1 that, in general, the "minimum standards" of the field are easily fulfilled, but I think we could still do better.

Reviewer 1 ·

Basic reporting

1) I could not find where the authors have made available the intermediate results, such as the codes and other data. As it is presented, the "chain of evidence" for interpretive purposes is not very explicit. I understand that is the case for many IS research papers, but I would like to rise the bar and have a bit more concrete description of the intermediate results during the analysis/interpretation stage.

2) Some terms should be clarified, and some expressions need to be re-formulated:
Examples:
Page 2, line 64: what you mean by "removed treatment"?
Page 3, line 123: please explain what you mean by "variance-based"
Page 3, line 125: "Our theoretical framework was primarily *based* on.."
Page 11, line 503: "*A* more tangible focus.."

Experimental design

I am very curious about the four-staged coding process. In particular, I am missing some middle-level results when going from focused coding to axial coding. In particular, why are no examples of categories extracted via axial coding? you jump directly from 308 codes to 6 themes, which is a quite big jump. In addition, I am interested on if there was any cross-validation of the coding (in particular during axial coding) across multiple researchers, or if it was only one researcher who did the entire interpretation. Also, can you elaborate further on the difference between initial and focused coding? - maybe with some examples?

When it comes to conflicting results between the manikin questionnaire and the qualitative data (here I am assuming the open ended interviews, the regular meetings and the observational sessions), how and when did you performed the clarifications when conflicting answers were identified? - an example could be very illustrative.

Validity of the findings

A major aspect that I am not extremely convinced in the proposed theory is the distinction made between Affects and Attractors. I believe the authors need to present a better case to justify the need of having two separate constructs, which appear to be difficult to clearly separate.

Also, I would suggest the authors to touch more actively upon how the specifics particular to the software engineering context affect the applicability of more general theories explaining the relationship of affect and performance. Basically, a more active comparison between the existing, more general theories with respect to the theory presented is advisable.

In the limitations of the work, the authors need to discuss their choice of having only males in an inductive study. It is well known that gender plays a major role on outcomes and behavior related to affect, and they should at least discuss how their sample may have affect the dynamics of the project, and thus the narrative from which the theory is derived.

Additional comments

This paper is well written, provides a good line of argumentation in relation to the relevance of the topic, it presents a sound methodology, and the results are presented in a clear, concise manner. The implications of affect on programming performance is a topic that it has caught up attention recently in SE, and the authors make a clear case for the "knowledge gap" they are trying to address. It was a pleasent experience to review this paper.

Reviewer 2 ·

Basic reporting

This paper presents a theory that explains the affects that software developers have and how these affects impact their performance while programming. To come up with the theory, the authors conducted a qualitative study and collected data through observations and various kinds of interviews. Using grounded theory techniques, the authors then established an explanatory theory.

The authors do a very good job in motivating why their research and in particular a theory of the impact of affects on software developers is necessary. In the paper, it is clearly explained what the implications of such a theory are, why a better understanding is necessary, and how the theory can help to get a better understanding. The authors also provide reasons why such a theory was not yet established, and why in general researchers have not yet focused much on software developers’ affects. Furthermore, the authors also put a lot of effort into explaining the theoretical background of their theory construction approach.

The paper is also well structured and written in a way that makes it easy to follow. However, in some cases, important details and explanations are missing. For example, the authors state that there were three discrepancies between the qualitative data and the SAM data, but all of these three discrepancies could be resolved. It would be interesting to know how these discrepancies were resolved and why there were discrepancies in the first place. Another aspect that needs clarification is the questionnaires that were used directly after a developer made a commit. What was the reason to let the participants fill out a questionnaire exactly at this point in time and what happened if a developer could not make any commit for a whole week (as mentioned in quote 16)? Didn’t that lead to a loss of data for this participant? How did the authors handle this situation? Furthermore, for me it is unclear how the paragraph from line 379 to 391 fits into the section about developers’ affects.

Experimental design

The experimental design that the authors have chosen is a good fit for their research. It allows them to observe developers during their work, and using grounded theory techniques, come up with a theory that explains the observations. However, there are two issues with the experimental design.

The authors should clarify what exactly they consider to be an emotion. There is already a lot of research in this direction, and different authors have a different opinion on how to distinguish between emotions, feelings, moods, etc. The authors state that in their work, they use the term affect to include emotions and moods and that’s fine, but I doubt that, for example, “feeling motivated”, or “feeling alone” would be considered neither an emotion nor mood by a lot of researchers in this area.

The other issue is about the sample size of the study and the environment in which the study took place. The authors state that theorist argue that theories should be simple in the beginning and by studying only two developers, they could keep the complex situation simple. Still, I’m a little bit concerned about the very small sample size and that the study took place in a separated work space. The two study participants did not interact with other developers, neither formally nor informally.

In general, the authors describe their experimental design in a very detailed why that would make it possible for other researchers to reproduce it. However, there are also some minor open questions that would benefit from clarification. First, it would be interesting to know whether the interviews were conducted in English or if the authors translated the participants’ answers. Second, the authors state that they conducted 5 impromptu meetings. Could the authors clarify which events or circumstances lead to calling an impromptu meeting? Third, for the affects, the authors performed a quantitative triangulation. Was there also such a triangulation done for the performance? If not, why? Fourth, in the section “Theory construction and representation”, the authors discuss the issues that come with the development of a theory and state that their study design has taken these issues into account. Could the authors elaborate how exactly their study design has tackled these issues?

Validity of the findings

The findings that the authors present are interesting and appropriately stated. The authors also discuss some implications of their findings for research but also for practitioners.

However, there are also some open questions that the authors should address. First of all, the authors state that affects and focus are in general positively correlated. Did the authors consider that there might also be negative affects that have a positive influence on the focus? For example, a developer might get very frustrated because s/he can’t fix a difficult bug, so that s/he puts a lot of effort in it, leading to a state of high focus that eventually helps to solve the difficult problem. Similarly, in Figure 1, it looks like events always trigger affects, but couldn’t it also be the other way round?

Furthermore, the category “Interventions” in Figure 1 seems to be very specific to this particular study setting, where a researcher could intervene when two bachelor students working on a software project couldn’t cope with each other. What would be the real world analog?

Additional comments

I only have two minor comments: i) in the enumeration starting on line 158, number 3) is missing, and ii) the authors grouped affects in two different categories: positive and negative affects. Later they refer to these two categories as two different dimensions of affects. Is this really correct? For me it looks more like two different values or categories of affects. Wouldn’t dimensions of affects be something like valence or arousal, i.e. different dimensions to measure an affect?

---

## Round 0.2 · accepted · Accept

I am happy to recommend this excellent paper for publication. However, maybe the authors want to run another round of proof-reading, as there are still a small number of typos in the manuscript (e.g., Line 39 "o software", or Line 47 "?.").